# A GENERAL DIFFERENTIALLY PRIVATE LEARNING FRAMEWORK FOR DECENTRALIZED DATA

## ABSTRACT

Decentralized consensus learning has been hugely successful that minimizing a finite sum of expected objectives over a network of agents. However, the local communication across neighbouring agents in the network may lead to the leakage of private information. To address this challenge, we propose a general differentially private (DP) learning framework that is applicable to direct and indirect communication networks without a central coordinator. We show that the proposed algorithm retains the performance guarantee in terms of generalization and finite sample performance. We investigate the impact of local privacy-preserving computation on the global DP guarantee. Further, we extend the discussion by adopting a new class of noise-adding DP mechanisms based on generalized Gaussian distributions to improve the utility-privacy trade-offs. Our numerical results demonstrate the effectiveness of our algorithm and its better performance over the state-of-the-art baseline methods in various decentralized settings.

## 1 INTRODUCTION

Decentralized learning is a process of learning a consensus model using the datasets that are distributed across different agents, such as machines, hospitals, and mobile devices (Shi et al., 2014; Han et al., 2017; Gong et al., 2016; Beyan et al., 2020). During the process, each local agent (1) keeps its own private data locally; (2) requires no exchange of raw data; and (3) communicates only with its connected agents to train its local model and updates the global parameters directly without a central coordinator. In particular, as medical data are inherently decentralized, i.e., owned or distributed across different institutions, direct sharing or central aggregation of such distributed medical data is increasingly restricted due to either ownership or other regulatory constraints. As a consequence, the advancement of decentralized learning will offer innovative solutions to transform healthcare sectors (Warnat-Herresthal et al., 2021).

Although decentralized learning only requires parallel computation at each local agent and sharing of the estimates or perhaps other intermediate parameters (auxiliary variables) with connected neighbouring agents, past experience has demonstrated the possibility of privacy leakage in the process: the attacker can still recover sensitive information from local communications (e.g., Fredrikson et al. 2015; Shokri et al. 2017). One defence procedure is to adopt a private variant of the learning algorithm using Differential Privacy (DP) to secure the training process. Very few DP algorithms focus on decentralized learning systems, with the exception of recent works in Xu et al. (2022); Yu et al. (2021a); Huang & Gong (2020). However, when introducing perturbation into the iterative learning process, these earlier methods only focus on achieving $(\epsilon, \delta)-$DP guarantee for each agent. Due to the communications with neighbouring agents during the iterative process, the overall privacy guarantee of the algorithm is no longer $(\epsilon, \delta)-$DP. Importantly, it is unclear how one can split the privacy budgets among all the agents in order to achieve a global $(\epsilon, \delta)-$DP guarantee for the algorithm when using these earlier methods. Finally, these existing methods consider a standard Gaussian noise-adding mechanism. The added unbounded noise could lead to unstable results, which can severely affect the learning efficiency and degrade the performance of the trained global model (Farokhi, 2022). This paper aims to provide a unified solution to address these issues and discuss the theoretical guarantees of the proposed algorithm.

## 1.1 RELATED WORKS

In the setting of centralized learning for distributed data, a handful number of papers have studied how to integrate privacy-preserving techniques, such as DP, into the training process (Jayaraman et al., 2018; Li et al., 2022a; Guo et al., 2020; Cai et al., 2018; Li et al., 2022b; Huang et al., 2020; Cao et al., 2021). For example, Jayaraman et al. (2018) proposed DP algorithms for convex problems where ensuring the information obtained from each local model satisfies DP guarantees before being aggregated in a central coordinator; Li et al. (2022b) proposed a unified centralized learning framework to ensure DP guarantees for each local agent for a general class of non-convex problems. However, these algorithms cannot be directly adapted to solve our problem; they focus on the setting where there exists a central coordinator that is responsible for aggregating information obtained from each local agent.

There have been very few recent developments in decentralized learning algorithms with DP guarantees. We are aware of only three recent works in the literature. Among them, Huang & Gong (2020) is the first one; they proposed a DP Alternating Direction Method of Multipliers (ADMM) algorithm for a wide range of convex learning problems, where they perturb the objective function before solving the minimization associated with the local dataset at each local agent. More recently, Yu et al. (2021a) proposed a DP decentralized stochastic gradient descent (SGD)-based algorithm by perturbing the intermediate parameter updates at each local agent before communicating the perturbed parameter updates with its connected neighbouring agents; Xu et al. (2022) proposed a blockchain-enabled decentralized DP learning algorithm through gradient perturbation. However, these gradient-based methods impose restrictions on the objectives, such as smoothness, and therefore have limited application in broader settings. In contrast, we don't restrict ourselves to using gradient descent to find the minimums of the target objective functions. Instead, by using operator theory, we solve the optimization problem by defining a suitable operator or mapping such that the fixed points are the solutions to the original problem; in other words, we consider a broader generic problem of finding a fixed point of averaged iteration of a nonexpansive mapping. Under such an operator theoretical framework, the SGD and ADMM algorithms previously considered in Huang & Gong (2020); Yu et al. (2021a); Xu et al. (2022) can be considered special cases of our proposed generic algorithm.

## 1.2 OUR CONTRIBUTIONS

In this paper, we propose a general framework of a decentralized learning algorithm with DP guarantee, referring to Differentially Private decentralized Krasnosel'skiĭ–Mann iteration (DP-dKM). Our contributions are summarized as follows.

1. Built on the Krasnosel'skiĭ–Mann(KM) iteration (Krasnosel'skii, 1955; Mann, 1953), we propose a unified decentralized learning framework with an overall DP guarantee, which is applicable to all communication network diagrams and covers many existing optimization algorithms, including the previously considered ones in Xu et al. (2022); Yu et al. (2021a); Huang & Gong (2020), as special cases.

2. We obtain an upper bound of the global sensitivity of the intermediate updates until the fixed iteration step; by injecting enough noises calibrated according to this upper bound, we can achieve a global $(\epsilon, \delta)-$DP guarantee without worrying about splitting the total privacy costs among different agents.

3. To the best of our knowledge, we provide the first decentralized learning algorithm that can achieve any desired overall DP guarantee while existing works rely on local DP mechanisms and therefore have no control over the overall privacy cost beforehand. And this is the first work that provides theoretical guarantees on the generalization and finite sample performance of the proposed algorithm.

4. To further optimize the privacy and utility trade-offs, we propose a class of truncated generalized Gaussian noise-adding DP mechanisms, which allows one to achieve significantly higher utility under the same level of DP guarantee.

5. Empirically, we conduct comprehensive experiments to demonstrate that our approach outperforms prior works in various decentralized settings characterized by different communication network diagrams.

## 2 PROBLEM STATEMENT AND PRELIMINARIES

In this section, we first start with the problem set. We then present preliminaries about decentralized learning schemes as well as differential privacy. Throughout this paper, we denote the $\ell_1$, $\ell_2$, $\ell_\infty$ norm of $x \in \mathbb{R}^p$ as $\|x\|_1$, $\|x\|$ and $\|x\|_\infty$. For a function $l(x, \xi) : \mathbb{R}^p \times \Xi \to \mathbb{R}$, $\|l\|_\infty := \sup_{x,\xi} |l(x, \xi)|$. For a matrix, $A$ denotes $A^\top$, $\|A\|_{op}$, $\|A\|_F$ as its transpose, spectral norm, and Frobenius norm respectively. Given another matrix $B$, $A \succ B$ means that $A - B$ is positively defined, and $A \succeq B$ means $A - B$ is positive semidefinite.

**Problem formulation:** Consider a network with $M$ agents, each of which holds a dataset $\Xi_m = \{\xi_{i(m)}\}_{i=1}^{N_i}$, for $m = 1, \cdots, M$, where $N_i$ is the number of training samples in the dataset $\Xi_m$, $\xi_{i(m)} \in \mathbb{R}^p$ is the $i$-th sample stored in the $m$-th agent. We assume that data are evenly collected and each agent has an equal sample size of $N$ for ease of presentation. Our primary focus is on solving a stochastic decentralized optimization approximated by its corresponding empirical loss,

$$\widehat{L}(x) := \min_{x \in \mathbb{R}^p} \frac{1}{MN} \sum_{m=1}^{M} \sum_{i=1}^{N} l_m\left(x, \xi_{i(m)}\right), \quad \widehat{x} := \arg\min_{x \in \mathbb{R}^p} \frac{1}{MN} \sum_{m=1}^{M} \sum_{i=1}^{N} l_m\left(x, \xi_{i(m)}\right),$$

where $x$ is the target parameter, and $l_m(\cdot)$s are the objectives that measure the performance of the local models. Throughout, we assume the objectives are convex, closed, and proper (*c.c.p*) but not necessarily differentiable. The goal is to learn a globally optimal solution, referring to consensus parameter (Cao et al., 2021; Shi et al., 2014) $\bar{x} = \frac{1}{M} \sum_{m=1}^{M} x(m)$, on $M$ agents across a network diagram, where $x(m)$ is the solution of the local parameter on the $m$-th agent. Among the agents, the estimates per iterate are peer-to-peer without the existence of a central coordinator, and its connection is typically modeled as a graph, e.g. Figure 4 in the Appendix. We stress that each agent operates independently and the average is only taken in the last iteration.

**Communication graph:** We now formally define the mathematical concept of graphs to characterize the communication among the agents. We define the connected network by, $\mathcal{G} = (\mathcal{V}, \mathcal{E})$ with vertex set $\mathcal{V} = \{1, \ldots, M\}$ and edge set $\mathcal{E} \subseteq \mathcal{V} \times \mathcal{V}$. We denote $\mathcal{N}(m)$ as the neighbour set of agents $m$. Edge $(m, l) \in \mathcal{E}$ represents the interconnection between agent $m$ and its neighbors $l \in \mathcal{N}(m)$. The decentralized optimization is associated with a given network topology that can be formulated mathematically by a mixing matrix (Alghunaim et al., 2019; Ying et al., 2021) and its properties can be summarized as follows.

**Definition 1 (Mixing Matrix)** *For any given graph $\mathcal{G} = (\mathcal{V}, \mathcal{E})$, the mixing matrix $\mathbf{W} = [w_{m,l}] \in \mathbb{R}^{M \times M}$ is defined on the edge set $\mathcal{V}$ that satisfies: (1) if $m \neq l$ and $(m, l) \notin \mathcal{E}$, then $w_{ml} = 0$; otherwise, $w_{ml} > 0$; (2) $\mathbf{W} = \mathbf{W}^\top$; (3) $\text{null}\{\mathbb{I} - \mathbf{W}\} = \text{span}\{\mathbf{1}\}$; (4) $\mathbb{I} \succeq \mathbf{W} \succ -\mathbb{I}$.*

We remark that $\mathbf{W}$ is a double gossip matrix that characterizes the communication among the agents and the matrix is non-unique for a given graph (Ying et al., 2021; Sun et al., 2021). Let $\lambda := \max\{|\lambda_2|, |\lambda_M|\}$, where $\lambda_i$ denotes the $i$th largest eigenvalue of $\mathbf{W} \in \mathbb{R}^{M \times M}$. The spectral gap as $1 - \lambda$ measures the connectivity of gossip communications among these agents (Zhu et al., 2022). Definition 1 implies that $0 \leq \lambda < 1$. A larger value of $\lambda$ indicates less exchange communication among local agents.

The KM iteration, as a simple implementation and fast convergence method in practice, has a long history and has been the most useful method in modern computing including operator-splitting and alternating-direction methods (Wotao, 2019; Wang et al., 2022). It is the basic and one of the most popular iterative schemes for finding one fixed point of a nonexpansive operator. Specifically, the KM algorithm offers several advantages over traditional optimization methods (e.g., Newton-type algorithms and interior-point methods) (Davis & Yin, 2016; Ryu & Yin, 2022; Liang, 2016): the former can easily handle nonsmooth terms and abstract linear operators, requires only simple arithmetic operations and scales well with the dimension of the problem. The KM additionally applies a decomposition procedure in which the original problem is broken into subproblems that can easily be solved (Ryu & Boyd, 2016). The KM iteration is widely applied to centralized learning (Chraibi & Takáč, 2019; Saber Malekmohammadi, 2021; Malinovsky et al., 2020).However, it is still an open question to perform the KM iteration in a decentralized learning setting. We next fill this gap by proposing the decentralized KM iteration and presenting detailed schemes for the local agents.

**Definition 2 (Stochastic Decentralized Krasnosel'skiĭ–Mann (dKM))** *Suppose the training sample set $\Xi := \bigcup_{m=1}^{M} \Xi_m$ is distributed-stored in $M$ agents with total sample size $NM$, where $\Xi_m$ is a training dataset located in the $m$-th agent for $m = 1, \cdots, M$. We assume that $\xi_{i(m)} \sim \mathbb{P}$ with $\xi_{i(m)} \in \Xi_m$ for any $m, i$. For each agent, given a nonexpansive operator $T$, the iterative formula of the stochastic dKM algorithm, $\mathcal{A}$, is defined as,*

$$x^{k+1}(m) = \mathcal{A}\left(x^k(m); \Xi\right) = \sum_{l \in \mathcal{N}(m)} w_{ml} x^k(l) + \alpha_k \left(T\left(x^k(m); \xi_{i_k(m)}\right) - x^k(m)\right), \quad (1)$$

*where $w_{ml}$ is the element of a given matrix $\mathbf{W}$ satisfying Definition 1, $\alpha_k \in (0, 1]$. $i_k$ is an i.i.d. variable drawn from the uniform distribution over $\{1, \cdots, N\}$ at the $k$-th iteration. Further, let $\mathbf{X} = [x(1), \cdots, x(M)]^\top \in \mathbb{R}^{M \times p}$ that stores all local parameters across the network, $\mathbf{T}(\mathbf{X}; \Xi) = [T(x(1); \Xi_1), \cdots, T(x(M); \Xi_M)]^\top \in \mathbb{R}^{M \times p}$ stacking all local updating w.r.t. the first argument. Iteration (1) has the matrix form, $\mathbf{X}^{k+1} = \mathbf{W}\mathbf{X}^k + \alpha_k \left(\mathbf{T}\left(\mathbf{X}^k; \Xi\right) - \mathbf{X}^k\right)$.*

As we consider the general framework of a decentralized learning problem with mild conditions (c.c.p.) for the loss, general computational procedures with wide coverage and flexibility that can be used to handle numerically inconvenient loss come more naturally. Specifically, the form of $T$ in Definition 2 depends on the specific algorithm we adopt. For example, dKM implies gradient descent, proximal gradient descent, and ADMM algorithms in a decentralized setting when choosing $T$ as a forward operator, forward-backward operator, and Douglas-Rachford operator. Please refer to Table 1 in the appendix for some forms of $T$. Additionally, the stochastic dKM algorithm provides a guideline to design a new decentralized learning algorithm by specifying the form of $T$.

**Privacy Concern:** Despite each agent communicating with its neighbours by sending parameters instead of directly exchanging raw data, the risk of leaking information still exists: the attacker can recover the sensitive information of data from shared parameters as discussed in Shokri et al. (2017), Fredrikson et al. (2015). This motivates us to consider privacy preserving iteration procedure with efficient communication while it retains a performance guarantee. Differential Privacy (DP), introduced by Dwork et al. (2006), is a widely adopted definition due to its important advantages over other privacy techniques. It quantifies to what extent individual privacy in a dataset is preserved while releasing aggregated information.

**Definition 3 (($\varepsilon, \delta$)-Differential Privacy Dwork et al. (2006))** *A stochastic algorithm $\mathcal{A}$ is called $(\varepsilon, \delta)$-differential privacy if for any subset $\mathbb{R}_0 \subset \mathbb{R}^p$ and any neighbouring sample set pair $\Xi$ and $\Xi'$ which differs by only one sample, we have $\log\left[\frac{\mathbb{P}_{\mathcal{A}(\Xi)}(\mathcal{A}(\Xi) \in \mathbb{R}_0) - \delta}{\mathbb{P}_{\mathcal{A}(\Xi')}(\mathcal{A}(\Xi') \in \mathbb{R}_0)}\right] \leq \varepsilon$.*

The common interpretation of $(\varepsilon, \delta)$-differential privacy is that it is $\varepsilon$-differential privacy except with probability $\delta$ (Mironov, 2017). The parameters $\varepsilon$ and $\delta$ are privacy budgets indicating the strength of privacy protection from the algorithm. The classic differential privacy is called $\varepsilon$-differential privacy with $\delta = 0$, which imposes an upper bound $e^\varepsilon$ on the multiplicative distance of probability distributions of randomized query outputs for any two neighbouring data sets (Dong et al., 2019).

## 3 SENSITIVITY OF THE STOCHASTIC dKM ITERATION

In this section, we estimate the $\ell_2$ norm sensitivity of the stochastic dKM, laying the foundation for noise addition in the truncated generalized Gaussian mechanisms in Section 5. Before formalizing the result, we present the assumptions throughout and introduce the definition of the sensitivity of algorithms in a decentralized learning setting.

**Assumption** *(1) The loss function is c.c.p. and sub-differentiable with respect to $x$, and the fixed-point iteration is bounded by a finite constant $B$, i.e., $\max_{x,\xi} \|T(x;\xi) - x\| \leq B$; (2) The loss function $l(x, \xi)$ is nonnegative and $\|l\|_\infty \leq R$ for some constant $R > 0$.*

$\|T(x;\xi) - x\|$ in Assumption (1) is defined as a fixed point residual in the literature which typically relates to the gradient of an objective function (Davis & Yin, 2016). We note that Assumption (1) is weaker than Yu et al. (2021a); Xu et al. (2022); Huang & Gong (2020); Sun et al. (2021); Zhu et al. (2022) as well as a common Assumption (2) in Sun et al. (2021); Zhu et al. (2022).

The sensitivity based on two datasets that differ at only one point is commonly used in Yu et al. (2021a); Xu et al. (2022); Huang & Gong (2020). Although the only different point stored at any local agent, the local communication among agents without a coordinator, affecting the full networks, promotes us to quantify its impact on a learning algorithm globally as introduced in Definition 4.

**Definition 4 (Sensitivity)** *For a specific algorithm $\mathcal{A}$ acting on training samples, $\Xi', \Xi''$ which are two adjacent datasets that differ by one data point. Until iteration $K$, define the $\Delta_K$-sensitivity of algorithm $\mathcal{A}$ as $\Delta_K := \sup_{\Xi', \Xi''} \|\mathcal{A}(\Xi') - \mathcal{A}(\Xi'')\|$.*

We are now establishing the $\Delta_K$-sensitivity of the dKM algorithm. That is, we, through Theorem 1, provide the boundedness on $\Delta_K$ due to the only one different point for any two adjacent datasets.

**Theorem 1 ($\Delta_K$-Sensitivity)** *Given $x^K = \frac{\sum_{m=1}^{M} x^K(m)}{M}$, $y^K = \frac{\sum_{m=1}^{M} y^K(m)}{M}$, denote $x^K$ and $y^K$ as the corresponding outputs of the dKM algorithm applied to two sets $\Xi'$, $\Xi''$ of size $NM$ which differ at only one point. Assume the initial value $\mathbf{X} = \mathbf{0}$. With Assumption (1) satisfied, given relaxed parameter, $\{\alpha_k\}_{k=0}^{K} \in (0, 1]$, the $\Delta_K$-sensitivity of the dKM algorithm has the upper bound,*

$$\mathbb{E}\Delta_K \leq \frac{2B \sum_{k=0}^{K-1} \alpha_k}{NM} + 4B \sum_{k=0}^{K-1} (1 + 2\alpha_k) \sum_{j=0}^{k-1} \alpha_j \lambda^{k-1-j}.$$

Note that the derivation of sensitivity of our proposed dKM algorithm does not require the assumption of smoothness and strong convexity of objective functions. Theorem 1 quantifies the accumulated deviation bound between two trajectories of iterates based on two datasets that differ at only one point, where it allows to exist at any local agent. Compared with Huang & Gong (2020) studying the local sensitivity, Theorem 1 establishes the global sensitivity as the local communication of the network makes this different point, e.g. storing in agent 1, affects the final output. The expectation in Theorem 1 comes from the randomness of picking the different point to update the iterate in Definition 2. Specifically, we pick the only different point for two adjacent datasets with probability $1/N$ to update the iterate and have $1 - 1/N$ chance using the same points. We, from Theorem 1, have that with a fixed iteration number $K$, as the data size, $M, N$ increases and $\lambda$ decreases, $\Delta_K$ gets smaller for both diminishing and constant learning rates. However, it fails to control the sensitivity when $K$ increases, which also suggests the risk of privacy that, with the higher iterative step, it will be easier to identify the specific sample. Moreover, the sensitivity decreases as $\lambda$ decreases indicating the effect of different topologies on $\Delta_K$. Table 2 in the appendix summarizes it for clarity. This theorem also provides a rule to establish the adding mechanisms to guarantee DP in Section 5.

## 4 PERFORMANCE AND GENERALIZATION OF DECENTRALIZED LEARNING ALGORITHMS WITH DIFFERENTIAL PRIVACY

Existing DP schemes in decentralized learning typically rely on the perturbation of objective functions, and gradients, but are limited to iterates (Yu et al., 2021a; Xu et al., 2022; Huang & Gong, 2020). Such methods usually introduce extra noise that has privacy preservation. It is still hard to examine the privacy and performance trade-off in the generalization of DP algorithms (He et al., 2021). In this section, we establish a generalization error bound and a finite sample guarantee of decentralized learning algorithms when these algorithms satisfy differential privacy. These results illustrate the effectiveness of using dKM with any differentially private mechanism (Definition 3) in applications. We next proceed by quantifying the bound considering iterate independent noise addition mechanisms and computing the end-to-end differential privacy guarantee across $M$ agents over a network system.

Let $L(x) = \mathbb{E}_{\xi \sim \mathbb{P}}[l(x, \xi)]$ and $x^\star$ be its optimal solution. Note that, for a specific stochastic algorithm $\mathcal{B} := (\mathcal{A}_1, \cdots, \mathcal{A}_M)$ on $\Xi$ with sample size $NM$ with output $\mathcal{B}(\Xi)$, where $\mathcal{A}_1, \ldots, \mathcal{A}_M$ performing on local agent allows being different, the excess generalization error of $\mathcal{B}$ defined as, $\mathbb{E}_{\Xi, \mathcal{B}}[L(\mathcal{B}(\Xi)) - L(x^\star)]$, can be decomposed into three terms (Bottou & Bousquet, 2007),

$$\underbrace{\mathbb{E}_{\Xi, \mathcal{B}}\left[L(\mathcal{B}(\Xi)) - \widehat{L}(\mathcal{B}(\Xi))\right]}_{\text{generalization error}} + \underbrace{\mathbb{E}_{\Xi, \mathcal{B}}\left[\widehat{L}(\mathcal{B}(\Xi)) - \widehat{L}(\widehat{x})\right]}_{\text{optimization error}} + \underbrace{\mathbb{E}_{\Xi, \mathcal{B}}\left[\widehat{L}(\widehat{x}) - \widehat{L}(x^\star)\right]}_{\text{test error}}. \quad (2)$$

We establish the boundedness of generalization error in Theorem 2 that reflects joint effects caused by the data $\Xi$ and the algorithm.

**Theorem 2 (Generalization Bound)** *Assume that the decentralized learning algorithm $\mathcal{B} : \Xi \mapsto \mathbb{R}^p \times \{1, \cdots, M\}$ is $(\varepsilon, \delta)$-differentially private. Under Assumption (2), we have that,*

$$\left|\mathbb{E}_{\Xi \sim \mathbb{P}^{MN}, \mathcal{B}(\Xi)}\left[L(\mathcal{B}(\Xi)) - \widehat{L}(\mathcal{B}(\Xi))\right]\right| \leq (1 - e^{-\varepsilon})R + e^{-\varepsilon}M\delta.$$

**Theorem 3 (Finite Sample Guarantee)** *Under the Assumption of Theorem 2, we have,*

$$\mathbb{P}(L(\mathcal{B}(\Xi)) \leq \widehat{L}(\mathcal{B}(\Xi)) + \epsilon) \geq \frac{\epsilon - (1 - e^{-\varepsilon})R - e^{-\varepsilon}M\delta}{\epsilon + R}, \text{ for any } \epsilon > 0.$$

These two theorems represent the gap between the empirical loss based on finite samples and its expectation. It demonstrates the impact of differential privacy on out-of-sample performance by establishing the bound of $L(\mathcal{B}(\Xi))$ in probability and expectation. Although ensuring data privacy sacrifices the generalization, these results show that a good privacy-preserving mechanism still retains a certain level of generalization as well as a finite sample guarantee. In the existing work for DP decentralized learning, Xu et al. (2022) provided convergence and regret analysis based on gradient aggregation and Gaussian mechanism in the presence of Byzantine nodes; Yu et al. (2021a) explored the convergence rate of DP-SGD algorithm with Gaussian mechanism; Huang & Gong (2020) theoretically analyzed the utility of DP-ADMM algorithm, which can be measured by the expected empirical risk with feasibility violation. Note that our theoretical results in Theorem 2 and 3 are suitable for all DP mechanisms. As far as we know, we are the first to establish a generalization bound and finite sample guarantee in DP decentralized setting.

We address that Theorem 2 and 3 require the algorithm $\mathcal{B}$ globally being $(\varepsilon, \delta)$-DP. Additionally, considering that each agent acts independently in practice, where there is less likely to reach an agreement on a consistent $(\varepsilon, \delta)$-DP across all agents (Bellet et al., 2018), we then proceed by investigating how the local computation would affect global differential privacy as a composition theorem which also provides the reasonableness of DP assumption in Theorem 2 and 3. In detail, Theorem 4 shows the level of overall privacy cost, given the privacy cost levels of the local agents. Similar results are discussed in Huang & Gong (2020); Yu et al. (2021a).

**Theorem 4 (Composition Theorem)** *Define iterates, which is similar to the output in Definition 2, generated by the specific stochastic algorithm with $K$ steps as $\{x^k\}_{k=1}^K$. For the $m$-th agent, denote $\tilde{\mathcal{A}}_m : \Xi \mapsto \{\tilde{x}^k(m)\}_{k=1}^K$, where $\tilde{x}^k(m)$ is the iterates corrupted by noise. For any fixed $m$, if $\tilde{\mathcal{A}}_m$ is $(\varepsilon_m, \delta_m)$-differential private, then $\tilde{X}^k = (\tilde{x}^k(1), \cdots, \tilde{x}^k(M))^T$ is $(\varepsilon', \delta')$-differential private, where,*

$$\varepsilon' = \min\{\varepsilon_1, \varepsilon_2, \varepsilon_3\}, \quad \delta' = 1 - \left\{\prod_{m=1}^M (1 - e^{a_m}\frac{\delta_m}{1 + e^{\varepsilon_m}})\right\} + \left\{1 - \prod_{m=1}^M (1 - \frac{\delta_m}{1 + e^{\varepsilon_m}})\right\},$$

*with, $\varepsilon_1 = \sum_{m=1}^M \varepsilon_m$, $\varepsilon_2 = \sum_{m=1}^M \frac{(e^{\varepsilon_m}-1)\varepsilon_m}{e^{\varepsilon_m}+1} + \sqrt{\sum_{m=1}^M 2\varepsilon_m^2 \log\left(e + \frac{\sqrt{\sum_{m=1}^M \varepsilon_m^2}}{\tilde{\delta}}\right)}$, $\varepsilon_3 = \sum_{m=1}^M C_{KL}(m) + \sqrt{2\log(\frac{1}{\delta'})(\sum_{m=1}^M \varepsilon_m^2)}$ with $C_{KL}(m) := \min\{\min\{2, e^{\varepsilon_m} - 1\}\varepsilon_m, \varepsilon_m\}$, for some $0 < a_m \leq \varepsilon_m$, $\sum_{m=1}^M a_m = \varepsilon'$, and real constant $\tilde{\delta}$.*

For completeness, Algorithm 1 shows the detailed iterative step of dKM with noise addition to preserve DP, and we further examine the optimization error bound in formula (2) that is caused by adding noise to the query output. Specifically, we consider iterate independent noise addition mechanisms (Definition 5) to preserve DP for dKM in practice: a random noise is added to the iterate to reduce leakage information.

**Definition 5 (Noise-adding Mechanisms for dKM)** *Given a data set $\Xi$, a query-output independent noise-adding mechanism $\tilde{\mathcal{A}}$ will release the query output $\tilde{x}^k = \tilde{\mathcal{A}}(x^k; \Xi)$ corrupted by an additive random noise $d$, $\tilde{x}^k = x^k + d$.*

---

**Algorithm 1** Differentially Private Decentralized Krasnosel'skiĭ–Mann Iteration (DP-dKM)

---

1: **Initialize**: $\widetilde{\mathbf{X}}^0$, mixing matrix $\mathbf{W}$, $\alpha_k \in (0,1]$, number of iterations $K$
2: **while** $k \leq K$ **do**
3:    **for** $m \in \mathcal{V}$ ($m \in [1, M]$) **do**
4:       $x^k(m) = \mathbf{W}\widetilde{\mathbf{X}}^{k-1}(m) + \alpha_{k-1}(T(x^{k-1}(m)) - x^{k-1}(m))$ (Local computation)
5:    **end for**
6:    **for** $m \in \mathcal{V}$ **do**
7:       Generate random noise $\varepsilon_m^k$, $\tilde{x}^k(m) = x^k(m) + \varepsilon_m^k$ (Differential Privacy)
8:       Broadcast $\tilde{x}^k(m)$ to all neighbours $j \in \mathcal{N}(m)$
9:    **end for**
10: **end while**
11: **Output**: $\mathbf{X}^K = (x^K(1), \cdots, x^K(M))$ and $\bar{x}^K = \frac{1}{M}\sum_{m=1}^M x^K(m)$

---

Let $\mathbf{X}^\star = [x^\star, \cdots, x^\star]^\top \in \mathbb{R}^{M \times p}$ be the true parameter and $\widetilde{\mathbf{X}} = [\tilde{x}(1), \cdots, \tilde{x}(M)]^\top$ be the released iterates corrupted by an additive random noise for each agent, the DP-dKM can be written as $\mathbf{X}^{k+1} = \mathbf{W}\widetilde{\mathbf{X}}^k + \alpha_k \left(\mathbf{T}\left(\mathbf{X}^k; \Xi\right) - \mathbf{X}^k\right)$. According to Assumption (1) and (2), the error bound is controlled by

$$\|\mathbf{W}\widetilde{\mathbf{X}}^k + \alpha_k \left(\mathbf{T}\left(\mathbf{X}^k; \Xi\right) - \mathbf{X}^k\right) - \mathbf{X}^\star\| \leq \|\mathbf{X}^{k+1} - \mathbf{X}^\star\| + \|\mathbf{W}\left(\mathbf{X}^k - \widetilde{\mathbf{X}}^k\right)\|,$$

where the first term is the same as in the non-privacy setting, which depends on the convergence properties of a given algorithm (Wotao, 2019). The second term indicates the deviation by using the privacy mechanisms. The following theorem gives the upper bound of the second term.

**Theorem 5 (Boundedness of local iterates based on Gaussian additive noise)** *Given a data set $\Xi$, assume an iterative independently noise-adding mechanism $\tilde{\mathcal{A}}$ releases the output $\tilde{\mathcal{A}}(x^k; \Xi) := x^k + d$ corrupted by an additive random noise $d$, where $d$ follows a Gaussian distribution with mean $\mu$ and variance $\sigma^2$. The error bound caused by the additive noise is,*

$$\mathbb{E}\left[\left\|\mathbf{W}\left(\mathbf{X}^k - \widetilde{\mathbf{X}}^k\right)\right\|_F^2\right] \leq p\left(\sigma^2 + \mu^2\right)\left[(M-1)\lambda^2 + 1\right].$$

# 5 DIFFERENTIAL PRIVACY VIA TRUNCATED GENERALIZED GAUSSIAN MECHANISMS

While commonly adopted Gaussian noise-adding mechanisms for a single iterate can guarantee DP (Croft et al., 2022; Cormode et al., 2019; Yu et al., 2021a; Xu et al., 2022; Huang & Gong, 2020), such mechanisms do not take into consideration the valid range of the iterates being posed and the utility of learning algorithm Geng et al. (2018); Bun et al. (2018): the extremely large noise will severely affect a learning process and degrade the performance of the trained model under differential privacy guarantee. For example, Yu et al. (2021a;b) gave the lower bound of variance of Gaussian noise to guarantee DP. Ganesh & Zhao (2020) considered $(\varepsilon, \delta)$-differential privacy with generalized Gaussian mechanisms to answer $k$ counting queries about a database. Different from Ganesh & Zhao (2020), our proposed generalized Gaussian mechanisms are novel due to the boundedness of noise. Specifically, we truncate the probability density function used for the generation of noise with a careful determination of an appropriate bounding parameter and propose truncated Generalized Gaussian (GG) distribution $\mathcal{P}_d := \text{GG}(0, \sigma, b)$ with location parameter 0, scale parameter $\sigma > 0$, shape parameter $b > 0$. Its probability density function is,

$$p(z \mid 0, \sigma, b) = C_{gg} \exp\left\{-\left(\frac{|z|}{\sigma}\right)^b\right\}, \text{ where } z \in [-A, A], \tag{3}$$

where $C_{gg}$ is a constant to guarantee $\int_{-A}^A p(z \mid 0, \sigma, b)dz = 1$. In the experiment, we use truncated GG noise with $b = 1, 2$, which represents the truncated Laplace distribution (Definition 6) and truncated normal distribution (Theorem 6) to preserve DP.

**Definition 6 (Truncated Laplacian Distribution Geng et al. (2018))** *Given the privacy parameters,* $0 < \delta < \frac{1}{2}$, $\varepsilon > 0$, *and iterates sensitivity* $\Delta > 0$, *the truncated Laplacian distribution with* $p = 1$ *in formula* (3) *preserves* $(\varepsilon, \delta)$-*differential privacy when taking* $\lambda := \frac{\Delta}{\varepsilon}$, $C_{Lap} := \left(2\lambda\left(1 - e^{-\frac{A}{\lambda}}\right)\right)^{-1}$, $A := \frac{\Delta}{\varepsilon}\log\left(1 + \frac{e^{\varepsilon}-1}{2\delta}\right)$.

**Theorem 6 (Truncated Gaussian Distribution)** *The truncated Gaussian distribution* $p_{nor}(z)$ *with* $p = 2$ *in formula* (3) *preserves* $(\varepsilon, \delta)$-*differential privacy, where* $\sigma^2 \geq \varepsilon^{-1}\Delta^2$, *the constants* $C_{nor}$ *and* $A$ *are determined by* $\int_{-A}^{A} p_{nor}(z)dz = 1$ *satisfying the equation,*

$$C_{nor} \cdot \sum_{l=0}^{\infty}(-1)^l \cdot \frac{A^{2l+1}}{\sigma^{2l}l!(2l+1)} = \frac{1}{2}, \quad C_{nor} \cdot \sum_{l=0}^{\infty}(-1)^l\frac{A^{2l+1} - (A-\Delta)^{2l+1}}{\sigma^{2l}l!(2l+1)} = \delta.$$

Note that the truncated Gaussian mechanism is also considered in Cesar & Rogers (2021) which focused on exploring privacy loss composition bounds for special classes of differentially private algorithms, while we aim to reduce the amount of added noise with the same level of privacy. An important property of the truncated GG mechanism is that the range of addition noises is bounded to $[-A, A]$ while the DP still holds. More importantly, the truncated GG mechanism simultaneously improves the utility and guarantees privacy. Its good performance compared with the state-of-the-art methods is illustrated in the numerical experiments.

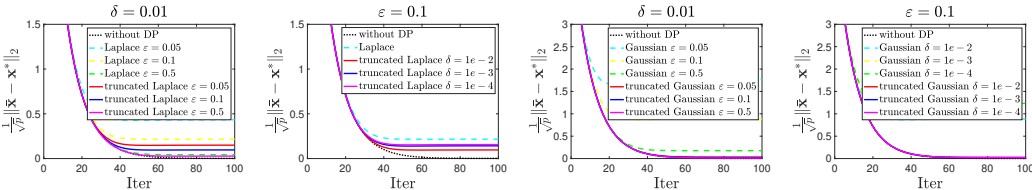

Figure 1: The sensitivity analysis of parameter $(\varepsilon, \delta)$ for SGD with a full connected graph.

## 6 NUMERICAL EXPERIMENT

We compare the proposed DP-dKM algorithm with baseline algorithms under various decentralized settings (ring, star, and full-connected graphs): (a) non-private decentralized approach; (b) private decentralized approach with Laplace noise; (c) private decentralized approach with Gaussian noise (Yu et al., 2021a; Huang & Gong, 2020). The average root means squared error (RMSE) is used to quantify their performance. To start with, we do a sensitivity analysis caused by privacy parameters $(\varepsilon, \delta)$ by solving least squares using the SGD algorithm on a fully connected graph. The results are shown in Figure 1.

In Figure 1, the first and second columns compare the performance of Laplace and truncated Laplace mechanisms with different $\varepsilon$ and $\delta$. Similarly, the comparisons between Gaussian and truncated Gaussian mechanisms are shown in the third and fourth columns. Figure 1 indicates that the proposed mechanism has the smallest RMSE compared with Laplace and Gaussian mechanisms and enjoys better convergence properties. In addition, the results demonstrate the privacy-utility trade-offs of the proposed approach: the RMSE increases as $\varepsilon$ increases with fixed $\delta$. When privacy leakage increases, the truncated Laplace and truncated Gaussian approach achieves better utility.

We next consider $\ell_1$ regularized least square regression and $\ell_1$ regularized logistic regression by employing the differentially private SPGD and ADMM algorithms with truncated generalized GG noise with $b = 1, 2$ to evaluate the performance of Algorithm 1,

$$\frac{1}{MN}\sum_{m=1}^{M}\sum_{i=1}^{N}(A_{mi}x - b_{mi})^2 + \lambda\|x\|_1, \quad \frac{1}{MN}\sum_{m=1}^{M}\sum_{i=1}^{N}\{\log(1 + e^{A_{mi}x}) - b_{mi}A_{mi}x\} + \lambda\|x\|_1.$$

The element $A_{mi}$, $x$ are drawn independently from the normal distribution. $\lambda > 0$ is a regularized parameter controlling the impact of the regularizer and is chosen by the grid search method. We fix

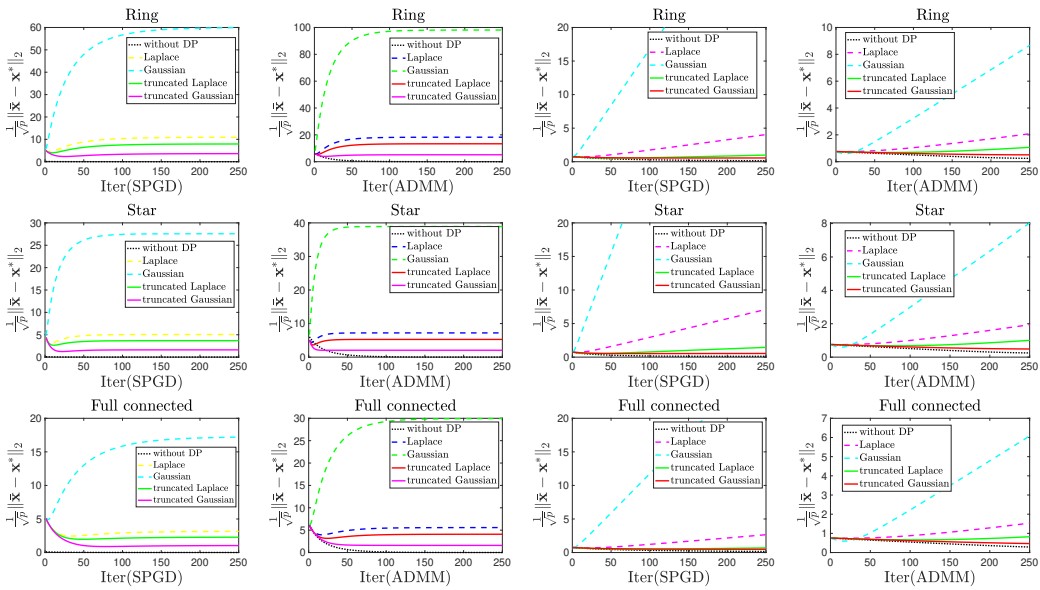

Figure 2: Estimation error on Ring, Star, and Full connected graphs. The first and two columns are $\ell_1$ penalized least squares; The third and fourth columns are $\ell_1$ penalized logistic regression.

the privacy budget $\varepsilon = 0.5$ and $\delta = 10^{-3}$ to evaluate the performance of the truncated approaches under the settings with different numbers of distributed data sources and different typologies. The results are summarized in Figure 2-3. We consider $M = 5, N = 100$ in Figure 2. Figure 3 explores the estimation error as the number of agents changes, showing that RMSE increases as agents increases. It also demonstrates that the truncated GG mechanism has the smallest RMSE than Laplace and Gaussian mechanisms, and keeps the same level of privacy. These results also show the effect of the topology graph on the convergence of the Algorithm 1: the algorithm converges faster with the star and full-connected graphs than the ring graph.

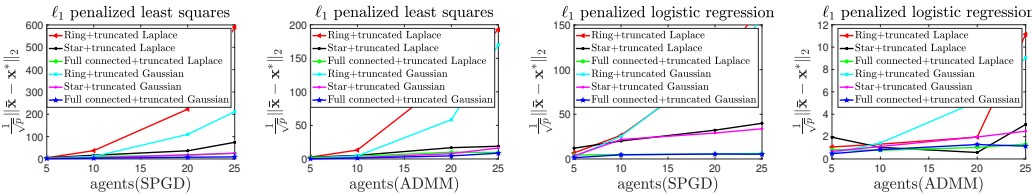

Figure 3: The estimation error as the number of agents changes.

# 7 CONCLUSION

In this paper, we have proposed a general framework of the privacy-preserving algorithm, DP-dKM, that is applicable to all communication network diagrams and covers many existing decentralized learning and optimization problems and show that the proposed algorithm retains the performance guarantee on generalization, and finite sample performance. We also established the effect of local privacy-preserving computation on global differential privacy. To avoid extremely large additional noise added to the shared information that will severely affect and degrade the performance of the learning process, we have introduced a truncated generalized Gaussian mechanism, in which we demonstrate privacy and utility trade-offs under a differential privacy guarantee. Experiments have demonstrated that our algorithm is effective in decentralized settings and performs better than the state-of-the-art baseline algorithms.

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
