# OpenReview forum: "A general differentially private learning framework for decentralized data"
_ICLR.cc/2023/Conference — Submitted to ICLR 2023_

### Official Review · Reviewer_HapE · 2022-10-23

**Confidence:** 3
**Correctness:** 3
**Technical Novelty And Significance:** 3
**Empirical Novelty And Significance:** 2
**Recommendation:** 3

**Clarity, Quality, Novelty And Reproducibility:**

Clarity: The paper is well organized but the presentation has minor details that could be improved.

Quality: The paper has minor technical flaws. For example, the proof of a theorem has some fixable errors or the experimental evaluation is weak.

Novelty: The paper contributes some new ideas or represents incremental advances.

**Strength And Weaknesses:**

Strength:
The targeted problem is interesting. This paper is easy to follow.

Weakness:
1. Differential private decentralized learning has been widely studied. The work aims to provide a general framework by simply combining SGD and SPGD algorithms. The contribution seems to be incremental.
2. In terms of the theoretical study, can the authors provide a detailed analysis of optimization error with differential private mechanisms? What are the convergence rates of the algorithms with SGD, and SPGD, respectively?
3. We would expect a more comprehensive experimental study. For example, the current experiments only focus on least square regressions with/without l1 regularized term. Can the proposed algorithm apply to classification problems such as logistic regression? What is the effect of the proposed algorithms under different privacy levels (epsilon, delta)?
4. Also, the proposed algorithm involves some parameters. The authors should provide a sensitive study on these parameters.
5. The text in the figures is too small.
6. How about the support recovery of the regularized least square regression with an l1 penalty with a noise-adding mechanism? It seems that the noise-adding ruins the spare structure easily. Can the authors provide an analysis of the noise-adding mechanism on structure recovery in addition to generalization error?
7. In Theorem 5, what is the sequence {a_m}? It makes delta^{\prime} different.

**Summary Of The Paper:**

This paper provides a differential private learning framework for consensus learning over networks. Some theoretical results are established for the proposed algorithms. Several numerical studies are demonstrated.

**Summary Of The Review:**

This paper provides a differential private learning framework for decentralized learning. However, the contribution seems to be incremental.

---

> ### Author Response · Authors · 2022-11-19
> **Response for your comments**
>
> General. The work aims to provide a general framework by simply combining SGD and SPGD algorithms. The contribution seems to be incremental.
>
> Response: We apologize for not being able to convey our contributions clearly enough in our previous version. We have rewritten the majority of the paper. We hope it is clear in our revised paper that our contributions are not simply combining SGD and SPGD algorithms.
>
> 1. The detailed analysis of optimization error with DP mechanisms and  the convergence rates of SGD and SPGD.
>
> Response: The detailed analysis of optimization errors requests specific algorithms by defining a specific form of the operator, which is not something we want to justify in this paper as we mainly want to quantify the part of optimization error caused by noise addition. The convergence rate of DP-SGD and DP-SPGD has been viewed as inexact stochastic algorithms of SGD, and SPGD and they have been well-studied in the literature (Yin, 2019).
>
> 2. The comprehensive experimental study, utility of the proposed algorithm on classification problems, and the effect of the proposed algorithms under DP levels.
>
> Response: We have added $\ell_1$ regularized logistic regression using SPGD and ADMM algorithms with truncated Laplace and truncated Gaussian mechanisms and make the comparison with Huang and Gong (2020) in the revised paper.  We also explore the sensitivity analysis to evaluate the effect of extra parameters $\varepsilon$ and $\delta$ in the experiment.
>
> 3. Sensitive study on these parameters.
>
> Response: We add the results of the sensitivity study for privacy parameter $(\varepsilon,\delta)$ in the revised version.
>
> 4. Figures presentation.
>
> Response: We have made the texts in the figures bigger in the revised paper.
>
> 5. The support recovery of the $l_1$ regularized least square regression with a noise-adding mechanism.
>
> Response: We have to admit that it remains a challenge to design an efficient differential privacy algorithm to preserve high recovery of the sparse structure in the regularized regression under a decentralized learning setting. As far as we know, no existing work has provided the sparse structure recovery analysis in the presence of  differential privacy, which is a limitation not only for our paper but also for other works. We will consider this open question in our future work.
>
> 6. In Theorem 5, what is the sequence $\{a_m\}$? It makes $\delta^{\prime}$ different.
>
> Response: $a_m$ is any sequence that can satisfy the constraint: (1) $0< a_m <=\varepsilon_m$, and (2) the sum of them equals to $\varepsilon^\prime$. Please see Theorem 4 on Page 6.
>
> 7. This paper provides a differential private learning framework for decentralized learning. However, the contribution seems to be incremental.
>
> Response: We apologize that our previous version was not clear enough to convey our scientific message. We hope the revised paper is clear enough to convince you that our contributions are not just incremental. Here we summarize our main contributions:
>
> (1) Built on the KM iteration, we propose a unified decentralized learning framework with an overall DP guarantee, which is applicable to all communication network diagrams and covers many existing optimization algorithms, including the previously considered DP-SGD and DP-ADMM algorithms in Huang et al.(2020), Yu et al.(2021) and Xu et al.(2022), as special cases.
>
> (2) We obtain an upper bound of the global sensitivity of the intermediate updates until the fixed iteration step; by injecting enough noises calibrated according to this upper bound, we can achieve a global $(\epsilon, \delta)-$DP guarantee without worrying about splitting the total privacy costs among different agents.
>
> (3) We provide the first decentralized learning algorithm that can achieve any desired overall DP guarantee while existing works rely on local DP mechanisms and therefore have no control over the overall privacy cost beforehand. And this is the first work that provides theoretical guarantees on the sensitivity, generalization, and finite sample performance of the proposed algorithm.
>
> (4) To further optimize the privacy and utility trade-offs, we propose a class of truncated generalized Gaussian noise-adding DP mechanisms, which allows one to achieve significantly higher utility under the same level of DP guarantee.
>
> (5) Empirically, we conduct comprehensive experiments to demonstrate that our approach outperforms prior works in various decentralized settings characterized by different communication network diagrams.
>
> Reference
>
> Wotao Yin. Operator splitting methods for decentralized optimization.
>
> Xu et al. SPDL: A Blockchain-enabled secure and privacy-preserving decentralized learning system.
>
> Yu et al. Decentralized parallel sgd with privacy preservation in vehicular networks.
>
> Huang et al. Differentially private ADMM for convex distributed learning: improved accuracy via multi-step approximation.

---

### Official Review · Reviewer_9m8T · 2022-10-24

**Confidence:** 4
**Correctness:** 4
**Technical Novelty And Significance:** 2
**Empirical Novelty And Significance:** 1
**Recommendation:** 3

**Clarity, Quality, Novelty And Reproducibility:**

The novelty of this paper is limited, which directly adds noise to the stochastic Decentralized Krasnosel’skiˇı–Mann (D-KM) iteration.

**Strength And Weaknesses:**

Strength:
1. Achieving privacy in decentralized learning is interesting and important.
2. The paper gives general framework based on D-KM iteration.
3. Proposed a truncated generalized Gaussian mechanism with higher utility.

Weakness:
1. Lots of decentralized learning with DP works are not considered.  Please check the corresponding works in
Li, Zhize, Haoyu Zhao, Boyue Li, and Yuejie Chi. "SoteriaFL: A unified framework for private federated learning with communication compression." arXiv preprint arXiv:2206.09888 (2022).
2. Please provide the experiments comparisons with previous works. For example, In Figure 4, please compare the proposed methods with previous DP-ADMM algorithms to show the effectiveness.  The same things for Figure 2 and 3.
3. The motivation for proposing truncated generalized Gaussian mechanism is not clear. Instead of the advantage of $l_2$ sensitivity, what is the advantage of truncated generalized Gaussian when compared with Laplace noise? Also, here is a work also proposed truncated Gaussian mechanism. [1] Cesar, Mark, and Ryan Rogers. "Bounding, concentrating, and truncating: Unifying privacy loss composition for data analytics." In Algorithmic Learning Theory, pp. 421-457. PMLR, 2021.
4. Please also provide the generalization and optimization error bound with previous DP-SGD, DP-SPGD and DP-ADMM.
5. Please provide more experiments over large datasets and neural networks models.

**Summary Of The Paper:**

This paper proposes a general DP decentralized learning framework based on stochastic Decentralized Krasnosel’skiˇı–Mann (D-KM) iteration, which can represent the common first-order algorithms, like SGD, SPGD, and ADMM. Also, based on previous truncated Laplace mechanism, they proposed a truncated generalized Gaussian mechanism.

**Summary Of The Review:**

Based on the weaknesses in previous part, the paper needs more concrete comparisons with previous works from theoretical and experiments and insights.

---

> ### Author Response · Authors · 2022-11-19
> **Response for your comments**
>
> Thank you for your positive assessment.  We have rewritten the entire Introduction section and added more technical details, interpretation of the results, as well as motivation for the KM algorithm and truncated generalized Gaussian mechanism.
> 1. Lots of decentralized learning with DP works are not considered.
>
> Response: We focus on a different setting from Li et al. (2022); there is no central server in our paper. We have added more contextual comparisons about decentralized and centralized learning in the Introduction section and cited more relevant papers, including Li et al. (2022).
>
> 2. Please provide the experiment comparisons with previous works.
>
> Response: These prior approaches  all use unbounded mechanisms to ensure privacy, we compared our algorithms with the ones using unbounded Gaussian and Laplace mechanisms (see references) to show the effectiveness of our truncated Generalized Gaussian mechanisms, see Pages 8 and 9.
>
> 3. The motivation for truncated generalized Gaussian mechanism and related works.
>
> Response: We apologize for missing the motivation and thank you for pointing out the reference. Our goal in choosing a truncated generalized Gaussian mechanism is to bound the magnitude of the injected noises needed to achieve DP. Without bounding the distribution of the noise, it is possible to add extremely large noises to the iterations, leading to unstable results. The advantage of the proposed mechanisms is about limiting the maximum magnitude of noises that can be possibly injected. Therefore, comparing the results obtained from truncated Laplace noise with traditional Laplace noise is more meaningful. As demonstrated in our numerical experiments, we achieve much better utility with the same level of privacy guarantee by considering a truncated mechanism, i.e., better privacy-utility trade-offs than prior works. Cesar and Rogers (2021) focused on a different goal; they studied privacy loss composition bounds for DP algorithms (bounded, unbounded and concentrated), while our goal is to bound the magnitudes of the injected DP noises. We have added the discussion below Theorem 6 on Page 8.
>
> 4. The generalization and optimization error bound with previous DP algorithms.
>
> Response:  To the best of our knowledge, we are aware of only three recent works in decentralized learning with DP guarantee. Xu et al. (2022) provided convergence and regret analysis based on gradient aggregation and Gaussian mechanism; Yu et al. (2021) explored the convergence rate of DP-SGD algorithm with the Gaussian mechanism; and Huang et al. (2020) theoretically analyzed the utility of DP-ADMM algorithm with Gaussian mechanism. The results in Theorem 2 and 3 are suitable for all DP mechanisms. However, none of these existing works studied the generalization error bound. Our paper fills this gap by providing the generalization error bound under our proposed unified learning framework in Theorem 5. These previously studied SGD, SPGD and ADMM algorithms can be reformulated and solved by our generic algorithm. We add the discussion on Pages 6 and 7.
>
> 5. The experiments over large datasets and neural network.
>
> Response: Thank you for your suggestions. An important practical challenge that decentralized learning is attempting to tackle is the lack of enough sample sizes at any particular local agent, motivating the use of datasets stored at more agents in order to increase the total sample size when the sample size at any local agent is often limited. Given this consideration, we conducted the experiments by varying the number of agents while keeping the sample size at each local agent fixed, instead of having large sample sizes at local agents.  We also like your idea to consider a neural network model. However, due to its unique practical challenges to run a neural network model locally at each agent, we believe implementing neural network models in decentralized learning settings deserves more thorough investigations. Therefore we will consider extending our current framework to accommodate neural network models in future work.
>
> 6. The novelty of this paper is limited.
>
> Response: We have revised the introduction and added more details to explain and compare our results with existing works. We hope that our revised paper is clear enough to convey our contributions.
>
> 7. Concrete comparisons with previous works.
>
> Response: We have added concrete comparisons with the previous works and added more explanation for the theoretical results, and added the results of $\ell_1$ regularized logistic regression, as well as the sensitivity analysis to evaluate the effect of extra parameters $\varepsilon,\delta$.
>
> References
>
> Xu et al., SPDL: A Blockchain-enabled secure and privacy-preserving decentralized learning system.
>
> Yu et al., Decentralized parallel sgd with privacy preservation in vehicular networks.
>
> Huang et al., Differentially private ADMM for convex distributed learning: improved accuracy via multi-step approximation.

---

### Official Review · Reviewer_Xztu · 2022-10-25

**Confidence:** 2
**Correctness:** 2
**Technical Novelty And Significance:** 1
**Empirical Novelty And Significance:** 1
**Recommendation:** 3

**Clarity, Quality, Novelty And Reproducibility:**

I have a number of issues with the writing/clarity of this paper.
1. The major one -- the graphs are impossible to read and interpret because of the font and the size. How am I supposed to see anything here? Am I supposed to zoom in and squint to see what is going on in there by looking at blurred images and text? This will sound harsh, but I really am not willing to do that because I feel that it is the responsibility of the authors to convince the reviewer about the quality of their work. The text in Section 6 is not enough for me to be able to justify what you obtained in those plots either. Where is the discussion on your experimental results? I would recommend by starting off by improving this section to be able to actually convey what you achieved through this algorithmic framework.
2. More context of this high-level problem of decentralised consensus learning would be more useful to differentiate with other versions of distributed learning. As a reader, I would want to understand your contributions in terms of the problem being solved first. Yes, there is a section on the problem statement, but it's not really sufficient to distinguish from the other well-known versions of the problems under the same umbrella.
3. How are the assumptions you make about the loss function (or anything else) different from those in the prior work? My understanding was that this work was also supposed to weaken those, and obtain more a general framework.
4. Around Definition 2, it might be helpful to provide more context on this D-KM algorithm. The concepts and the ideas seem to be coming out of thin air right now. For example, what is the high-level idea of the bigger algorithm that uses this iterate? What algorithmic techniques are relevant for this particular design?
5. Also, is improving on the communication complexity one of the main highlights of this approach? The end of page 4 vaguely has a comment of this nature. If so, then how does it compare with that of the relevant prior work?
6. In Theorem 1, why just bound the expected sensitivity? How does this translate to a bound on the worst-case sensitivity? More context here would help! Also, the second summation should have $K$, instead of $k$.
7. The results and the settings are stated in very confusing ways, for example, in Theorem 5. I think the authors could do a better job of providing more context behind what the results are about and what they actually say. It's pretty hard to interpret the results the way they are right now.
8. The previous point brings me to my next complaint that the organisation and the flow of this paper are a bit confusing and haphazard. Sure, the order of the sections makes complete sense, but the ordering within them didn't feel as friendly. For example, the $\Delta_K$ sensitivity in Section 4 is defined, but its theorem is about its expected value, and then there is a table below about that for different graph topologies. What is the context here? Why do these graph topologies matter in particular? How are these three things connected? This was just one example. This may sound a bit vague, but the flow seems a bit jittery right now, in general, and maybe another iteration of revising it maybe helpful for future.
9. Some additional results about these generalised distributions (Laplace and Gaussian) could be more meaningful. For example, the privacy guarantees in terms of the extra parameters, or some tail-bounds. They don't add much value here in terms of clarity otherwise because I don't really know how they are useful for your DP framework.
10. Some theoretical comparisons with prior work could be useful.

In terms of the quality and the novelty, I don't really have much to say. I'm not very convinced about either of those because this paper wasn't really friendly to read or interpret.

**Strength And Weaknesses:**

Strength:
1. A new private framework is provided to deal with this problem that utilises generalised versions of distributions, like Laplace and Gaussian. There is mathematical rigour in this paper, which I appreciate.

Weaknesses:
1. One major complaint I have is about the writing quality of this paper. It pretty much makes it impossible for me to evaluate its significance and quality. I'll describe more in the next section.
2. I don't completely understand the objective or the motivation of this paper, to be honest. What assumptions are being made/removed about the loss functions, and why is this framework stronger and more general-purpose than in the prior work we have seen so far? What is it trying to improve on -- what is the prior work here? What are the baseline (trivial or non-trivial) approaches I could compare this work with? This point does tie in with the previous point I made here.
3. Theoretical comparisons with some prior work could have given more context.

**Summary Of The Paper:**

This paper develops a new differentially private (DP) framework to tackle the problem of decentralised consensus learning. This framework is based on the stochastic Decentralised Krasnosel'skii-Mann (D-KM) iteration (its privatised version referred to as "DP-KM" in this paper). It provides both theoretical and empirical results for their approach.

**Summary Of The Review:**

I don't think I got enough information from this paper to be able to provide a convincing argument in favour of accepting it. I hope my suggestions/comments on the writing help the authors.

Update: I appreciate and respect the authors' willingness to make all the suggested edits to improve the readability. However, I see that a *very significant* portion of the paper had to be rewritten in order to introduce more clarity, so I am hesitant to change my score. The readability has improved a lot, but I think the authors might benefit from spending some more time on the presentation of the paper, and providing a more convincing argument about the problem and the technical challenges and novelty.

---

> ### Author Response · Authors · 2022-11-19
> **Response for your comments**
>
> Thank you for your valuable suggestions and comments that have allowed us to greatly improve the clarity and flow of our paper. We hope that you find our revised paper satisfactory.
>
> 1.Graphs
>
> Response:We revised the graphs. We also rewrote the method section by adding more discussion of experimental results.
>
> 2.Problem statement
>
> Response:We rewrote the Introduction with more background about decentralized consensus learning,  the comparison with centralized learning, and more detailed comparisons with relevant works.
>
> 3.Assumptions
>
> Response:The loss function is assumed to be closed, convex, and proper, but not necessarily differentiable. In this sense, we do require weaker assumptions than the prior works in Xu(2022), Yu(2021), Huang(2020). We added it in the revised paper.
>
> 4.Motivation for D-KM algorithm
>
> Response:Under the operator theoretical framework, many existing algorithms (e.g., SGD, ADMM) can be formulated as finding a fixed point of averaged iteration of a nonexpansive mapping and can be solved through the well-established KM iteration. Our D-KM algorithm (now called dKM in revision) extended the existing KM algorithm to a decentralized learning setting. In this sense, our proposed algorithm provides a unified framework for solving many optimization problems in decentralized settings and covers the previously studied decentralized SGD and ADMM algorithms. We added this discussion in the Introduction and the paragraphs before and after Definition 2.
>
> 5.Communication complexity
>
> Response:The communication complexity, characterized by the parameter $\lambda$, is determined by the network structure of the agents, i.e., how they can communicate with neighbouring agents. The remark at the end of page 4 meant to compare our result stated in Theorem 1 with a similar result obtained previously for a centralized learning setting by Sun(2021). We want to point out the cost of studying a decentralized learning framework in comparison to a centralized counterpart: less algorithmic stability due to increased global sensitivity with an extra term given in Theorem 1 in comparison to the one obtained by Sun(2021).
>
> 6.Expected sensitivity
>
> Response:The expectation in Theorem 1 comes from the randomness of picking different data to update the iterate in Definition 1 due to the stochastic nature of our algorithm. The existing literature focuses on sensitivity that uses a fixed number of samples. However, the learning algorithms employed could be stochastic algorithms that draw a varying number of samples for updating per iteration. And, releasing their result may endanger privacy: the maximal deviation caused by changing one sample could be arbitrarily large. A similar issue was also discussed by Wang(2022), which shows that a standard DP mechanism with finite sensitivity fails to achieve DP. The expected sensitivity takes into account the randomness of the data sampling by the stochastic algorithm, which is similar to the one defined for expected differential privacy by Wang(2022). We refer to the upper bound as “worst case” sensitivity since this is not the exact value of the sensitivity and might be higher than required.
>
> 7.The context of the results
>
> Response:We have added more contexts to all results and reorganized theorems to improve the flow of our paper. This theorem provides a complementary result to further demonstrate the cost of the level of overall privacy for the entire learning process under our framework if one wants to impose agent-specific privacy levels. Similar results with the Gaussian mechanism are discussed by Huang(2020) and Xu(2021). However, these papers did not discuss how to properly choose a noise level for each of the local agents in order to achieve the desired global privacy guarantee for all DP mechanisms. Theorem 1 fills this gap by providing an upper bound of the global sensitivity, a quantity needed to determine the magnitude of the DP noises to achieve an overall $(\varepsilon, \delta)$−DP.
>
> 8.$\Delta_K$ sensitivity
>
> Response:The sensitivity decreases as $\lambda$ decreases indicating the effect of different topologies on $\Delta_K$. Numerical results explained the impact of topologies. We added more details to explain the definition.
>
> 9.Additional results about generalized distributions
>
> Response:We have added more experiments, including sensitivity analysis.
>
> 10.Comparisons with prior work
>
> Response:We addressed the difference with existing works in the revised version (highlighted in blue).
>
> References
>
> Xu. SPDL: A Blockchain-enabled secure and privacy-preserving decentralized learning system.
>
> Yu. Decentralized parallel SGD with privacy preservation in vehicular networks.
>
> Huang. Differentially private ADMM for convex distributed learning: improved accuracy via multi-step approximation.
>
> Sun. Stability and Generalization of Decentralized Stochastic Gradient Descent.
>
> Wang. Differentially Private Algorithms for Statistical Verification of Cyber-Physical Systems.

---

### Official Review · Reviewer_boyW · 2022-11-01

**Confidence:** 2
**Correctness:** 3
**Technical Novelty And Significance:** 2
**Empirical Novelty And Significance:** 2
**Recommendation:** 5

**Clarity, Quality, Novelty And Reproducibility:**

I believe the privacy proof for truncated generalized Gaussian is a novel contribution. For the rest of the contribution, it seems to me as more like following the familiar route of privatizing a known non-private algorithm.

**Strength And Weaknesses:**

The main idea in this paper seems to be to use the private version of stochastic D-KM algorithm. The paper analyzes its sensitivity and from there uses truncated generalized Gaussian mechanism for privacy. The paper claims privacy proof for truncated generalized Gaussian mechanism as well. I believe this can be casted as one of the novel contribution of the paper, too.

**Summary Of The Paper:**

The paper claims to give a general DP framework for decentralized data. The authors claim that the algorithm retains all the desirable properties that we have in non-private setting: generalization or stability and finite sample performance.

**Summary Of The Review:**

I was unable to read the proofs of the main claim, so I cannot make a judgement of their correctness. For now, I would rely that the authors have done their paper in writing the correct proof.

I have one central question. Ganesh and Zhao proved that generalized Gaussian is DP for all b \leq \log(k), where k is the number of queries (in their paper). The current submission makes similar statement but only for b=2. How is the proof different or anyhow technically novel considering the paper by Ganesh and Zhao (2020) and adapting their proof for truncated Gaussian just like Geng et al. did for truncated Laplacian? Am I missing something here? The authors have not cited Ganesh and Zhao, so I am wondering if the authors are aware of the paper.

Second, the authors claim that they present their result for equal sample size for the ease of presentation. Is it just for the ease of presentation or is it the case that the analysis only goes through when sample sizes are equal. I would imagine that if the sample size comes from a very skewed distribution, then we should not be able to say anything meaningful about the generalization error.

Why do we call the mean parameter vector a consensus vector? Is it common in this literature?

What is $\lambda$? I am guessing it is the rate of the convergence to the stationary distribution.

What is the notation $\| \ell \|_\infty$?

Theorem 5: What is $C_{KL}$ and is other $\widetilde X(i)$ defined similarly to $\widetilde X(1)$? Theorem statement should be precise and complete on their own.

The notation $\mathbb P$ is overloaded.

Disclaimer: I am trying to read the proof. I hope to be able to engage with the authors much more during the discussion phase. My current scores are for placeholder.




[1] Ganesh and Zhao. https://arxiv.org/abs/2010.01457

---

> ### Author Response · Authors · 2022-11-19
> **Response to your comments**
>
> We thank you for your valuable comments and suggestions. We have addressed your concerns in our revision. We hope you find our efforts satisfactory. Below we provide point-by-point responses to your comments.
>
> 1. I was unable to read the proofs of the main claim, so I cannot make a judgment of their correctness. For now, I would rely on the authors who have done their papers in writing the correct proof.
>
> Response: We have double-checked our proofs to make sure they are free of errors.
>
> 2. Comparison of proposed truncated Generalized Gaussian mechanism with existing work (Ganesh and Zhao (2020)).
>
> Response: The proposed truncated Generalized Gaussian mechanism can be extended to the case with $b \geq 3$. We highlight that the proof of our proposed generalized Gaussian mechanisms is completely different from that of Ganesh and Zhao (2020) mainly because of the boundedness of the tail, which play an important role in the proof or privacy guarantee. Specifically, due to the truncation nature of our DP mechanism, the probability of being outside of the chosen boundaries is exactly 0 rather than bounded by an exponentially vanishing bound as considered in Ganesh and Zhao (2020) (Lemma 9). In our proof, the major challenge is to determine the boundary parameter, $A$ in order to achieve the desired level of differential privacy guarantee, leading to some parameters, such as $\Delta$, $\sigma$, and $\varepsilon$ with fixed at certain values. However, when $b \geq 3$, this requires solving multiple equations or using the approximation methods thereof, complicating the computation of $A$. Therefore, we focus our attention on the solution when $b=1,2$, which has been shown in our numeric experiments to perform much better than prior works and importantly can be easily applied in practice due to its simplicity. A detailed discussion is given in the revised paper, see Page 7.
>
> 3. Question regarding equal sample sizes.
>
> Response: Yes, the use of an equal sample per agent is for ease of presentation. Our conclusion on the generalization bound can be easily extended to accommodate the setting of different sizes for different agents, where we replace the total sample size $NM$ by the sum of $N_m$.  The generalization upper bound only involves parameters $R,\varepsilon,\delta$ and the number of agents $M$, and therefore it does not change with different sample sizes in local agents.
>
> 4. Why do we call the mean parameter vector a consensus vector? Is it common in this literature?
>
> Response: Sorry about the confusion. In the decentralized optimization literature, it is ubiquitous to call the mean parameter vector (i.e., the common solution of the global model) a consensus vector, e.g. Shi et al.(2014), Cao et al. (2021). We added the explanation in the revised paper to avoid future confusion (see Page 3).
>
> 5. What is $\lambda$? I am guessing it is the rate of convergence to the stationary distribution.
>
> Response: $\lambda:=\max (\{|\lambda_2|,|\lambda_M|\} )$ measures the rate of communications among the agents, where $\lambda_i$ denotes the $ith$ largest eigenvalue of mixing matrix $\mathbf{W} \in {R}^{M \times M}$. A larger value of $\lambda$ indicates less exchange communication among local agents. We have added this interpretation of $\lambda$ in the paper on Page 3.
>
> 6. What is the notation $|l|_{\infty}$?
>
> Response: For a function $l(x,\xi): \mathbb{R}^p \times \Xi\rightarrow \mathbb{R}$, define $\|l\|_\infty:=\sup_{x,\xi}|l(x,\xi)|$. We have added this definition in the revised paper on Page 3.
>
> 7. Theorem 5: What is $C_{KL}$ and is other $\tilde{X}(i)$ defined similarly to $\tilde{X}(1)$? The theorem statement should be precise and complete on its own.
>
> Response: $C_{K L}(m):=\min (\{\min (\{2, e^{\varepsilon_m}-1)\} \varepsilon_m, \varepsilon_m\})$. We have added this definition in the revised paper. Also, we have fixed this notation confusion about $\tilde{X}(i)$ by introducing a new one, $\tilde{X}^k=(\tilde{x}^k(1),\cdots,\tilde{x}^k(M))^T,$ to replace $\tilde{X}(i)$. See page 6 in the revised paper.
>
> 8. The notation $\mathbb{P}$ is overloaded.
>
> Response: We have reduced the unnecessary notation for clarity in the revised paper.
>
> References
>
> Ganesh and  Zhao.  Privately Answering Counting Queries with Generalized Gaussian Mechanisms.
>
> She et al., On the linear convergence of the ADMM in decentralized consensus optimization.
>
> Cao et al., Differentially private ADMM for regularized consensus optimization.
>
>  Xu et al., SPDL: A Blockchain-enabled secure and privacy-preserving decentralized learning system.
>
>  You et al., Decentralized parallel SGD with privacy preservation in vehicular networks.
>
>  Huang et al., Differentially private ADMM for convex distributed learning: Improved accuracy via multi-step approximation.

---

> > ### Comment · Reviewer_boyW · 2022-12-07
> > **Response to rebuttal**
> >
> > I apologize that I have been very late in my response. Thanks for your response. A few clarifications might help me understand the paper better.
> >
> > 1. For comparison with Ganesh and Zhao: Can the authors quickly remind me what is the benefit of having a boundedness condition on the pdf? I might be missing something here. I want to understand this because I want to really understand the tangible advantage over GZ20 given that for generalized Gaussian as in GZ20. Does it mean that in this case, we get probability 1 bounds?
> > 5. So, in this sense $\lambda$ plays the role of the spectral gap which shows the expansion on the corresponding matrix.

---

> > > ### Author Response · Authors · 2022-12-07
> > > **Response to rebuttal**
> > >
> > > 1.For comparison with Ganesh and Zhao: Can the authors quickly remind me what is the benefit of having a boundedness condition on the pdf? I might be missing something here. I want to understand this because I want to really understand the tangible advantage over GZ20 given that for generalized Gaussian as in GZ20. Does it mean that in this case, we get probability 1 bounds?
> > >
> > > Response: Yes. The advantage of truncated generalized Gaussian noise (our work) is the boundedness guarantee of added noise with probability 1, while the added noise can be extremely large with a probability greater than 0 in GZ20's work.
> > >
> > > 2.So, in this sense $\lambda$ plays the role of the spectral gap which shows the expansion on the corresponding matrix.
> > >
> > > Response: Yes. A larger value of $\lambda$ indicates less exchange communication among local agents.

---

### Decision · Program_Chairs · 2023-01-20

**Decision:**

Reject

**Justification For Why Not Higher Score:**

The work is incremental with limited algorithmic novelty.

**Justification For Why Not Lower Score:**

N/A

**Metareview: Summary, Strengths And Weaknesses:**

The paper proposes a framework for differentially private (DP) learning for decentralized data based on stochastic Decentralized Krasnosel'skii-Mann iterative method. The central claim is that the proposed algorithm retains the performance guarantees in the non-private setting.

This work aims at developing algorithms for decentralized learning with provable privacy guarantees, which is an important research problem. The privacy guarantee is based on a truncated version of the generalized Gaussian mechanism. The mechanism is a tweaked version of an existing method.

The work appears to be mostly incremental compared to prior work and has limited conceptual and algorithmic novelty. The privacy analysis is largely similar to those in prior works (e.g., [Geng et al. 2018], [Ganesh & Zhao 2021], and [Cesar & Rogers 2021]) as pointed out by two reviewers. The authors claim that the analysis is different but the differences they claim appear to be simple tweaks. Also, the improvement in the guarantee on the added noise is not very significant.

Moreover, there are major concerns about the presentation and writing quality, which make the reader unable to follow the claims and assess their significance. This seems to apply to the formal statements, proofs, and also the experimental results.

The paper could also benefit from a more detailed discussion of related work on decentralized learning/optimization with DP including earlier works on locally DP learning.